# The Youngest, the Heaviest and/or the Darkest? Selection Potentialities and Determinants of Leadership in Canarian Dromedary Camels

**DOI:** 10.3390/ani11102886

**Published:** 2021-10-03

**Authors:** Carlos Iglesias Pastrana, Francisco Javier Navas González, Elena Ciani, Ander Arando Arbulu, Juan Vicente Delgado Bermejo

**Affiliations:** 1Department of Genetics, Faculty of Veterinary Sciences, University of Córdoba, 14014 Córdoba, Spain; ciglesiaspastrana@gmail.com (C.I.P.); juanviagr218@gmail.com (J.V.D.B.); 2Institute of Agricultural Research and Training (IFAPA), Alameda del Obispo, 14004 Córdoba, Spain; 3Department of Biosciences, Biotechnologies and Biopharmaceutics, Faculty of Veterinary Sciences, University of Bari ‘Aldo Moro’, 70121 Bari, Italy; elena.ciani1976.ec@gmail.com; 4Animal Breeding Consulting, University of Córdoba, 14071 Córdoba, Spain; anderarando@hotmail.com

**Keywords:** dromedary camel, sex-separated breeding, intraherd leadership, animal handling improvement, pleiotropic genes, morphofunctional selection

## Abstract

**Simple Summary:**

Genetic selection of camels for behavioral traits is not an extended practice in livestock scenarios. Given the existence of pleiotropic genes that influence two or more seemingly unrelated phenotypic traits, here we studied the sociodemographic, zoometric and phaneroptical characteristics potentially determining the intraherd leadership role in Canarian camels. This local endangered breed is mainly reared in same-sex groups because of biased morphostructural preferences, that is, tourism/leisure and milk production for males and females, respectively. The attribute most influencing leadership role was sexual status, as gelded animals more frequently initiated group movements. Furthermore, younger camels were mainly endorsed as group leaders, a condition that could be ascribed to their recognized fluid intelligence and need for constant social and environmental interaction. Referring to zoometrics and phaneroptics, the heaviest and darkest-coated dromedaries were significantly more prone to reaching higher positions in the leadership hierarchy. The presence of white-haired zones in the extremities, head and neck as well as iris depigmentation had non-negligible influence on this type of social organization. This information is valuable for application both in refining animal handling procedures and in genetic selection of animals for their social behavior.

**Abstract:**

Several idiosyncratic and genetically correlated traits are known to extensively influence leadership in both domestic and wild species. For minor livestock such as camels, however, this type of behavior remains loosely defined and approached only for sex-mixed herds. The interest in knowing those animal-dependent variables that make an individual more likely to emerge as a leader in a single-sex camel herd has its basis in the sex-separated breeding of Canarian dromedary camels for utilitarian purposes. By means of an ordinal logistic regression, it was found that younger, gelded animals may perform better when eliciting the joining of mates, assuming that they were castrated just before reaching sexual maturity and once they were initiated in the pertinent domestication protocol for their lifetime functionality. The higher the body weight, the significantly (*p* < 0.05) higher the score in the hierarchical rank when leading group movements, although this relationship appeared to be inverse for the other considered zoometric indexes. Camels with darker and substantially depigmented coats were also significantly (*p* < 0.05) found to be the main initiators. Routine intraherd management and leisure tourism will be thus improved in efficiency and security through the identification and selection of the best leader camels.

## 1. Introduction

Selective mimetism can be understood as the joining decisions made by individuals based on external factors such as proximity to other individuals. This form of behavioral coordination makes, for instance, gregarious animals more likely to engage in a movement when in close proximity of other congeners [1]. Selective mimetism may bidimensionally operate at both time and space levels across the members composing a group but always dynamically seeks the stability of a specific behavioral pattern [2]. 

In this context, the members of the social intraherd network may tend to follow one or more leader(s) [3], which are animals that more frequently initiate collective movements [4]. These animals are often preferably selected by herdsmen/conservationists in domestic/rewilding scenarios for the maintenance of group cohesion in farmland activities [5] and free-roaming herds [6]. Consequently, the study of the initiation and propagation of collective movements on the basis of sociodemographic attributes may not only permit better understanding of how species-specific social structure affects animal space use patterns, but also reveal breeding criteria to facilitate the labor of domestic gregarious animals’ handlers. 

As stated in [6], leadership is expected to be a complex, multifactorial interaction process, far from a strictly despotic action. Various characteristics have been indicated in literature to affect movement control in animal groups. For example, for some mammalian wild species, the oldest individual, who is supposed to better know the surrounding environment, may frequently act as the leader [7,8,9]. Contrastingly, although age has been reported not to condition leadership, it has been reported to be a mediator of dominance status in feral horses and zebras [10,11,12], brown lemurs [13] and macaques [14]. 

Among other conditioning factors, gender [15,16] and physiological individual state have frequently been shown to interact and be determinant for certain sex-dependent conditions. For instance, pregnant or lactating females have been suggested to lead the group towards areas where the resources for the satisfaction of their high energy requirements can be found [17]. For domestic species, extended knowledge on animal leadership behavior is currently available for dogs [18], swine [19] and other major ungulates [20,21]. However, sparse effective scientific knowledge exists within this applied field for rare, minor domestic species such as camels [22].

In regard to hierarchy determination and leadership, the most common observational findings in camel mixed-sex herds depict an older male in the lead who is followed by the rest of the congeners [23] and the internal division or large herds into smaller numerous subgroups [24]. However, the patterns of social relationships within single-sex camel herds have not been evaluated yet. 

Deepening the knowledge in this field may assist in defining handling standards adapted to the species’ social ecology. Furthermore, because of the gregariousness nature of camels [25] and the emerging socioeconomic interests in this species’ productive exploitation [26], it is imperative to be conscious of animal-dependent traits that differentially contribute to the establishment of intraherd social rank and how these can be used to make handling easier through individual selection. The immediate need for this approach is justified on account of the practice of sex-separated breeding for some local camel breeds (i.e., Canarian camels) for utilitarian purposes attending to sex-biased morphostructural preferences [27]. 

In the present research, we aimed to evaluate the relative importance of sociodemographic, zoometric and phaneroptical characteristics in determining the intraherd leadership role in Canarian camels. The deeper the knowledge on the occurrence and dynamics of leadership in camels, the better livestock management practices can be adjusted for animal welfare promotion and the maintenance of handlers’ self-security. More specifically, apropos of Canarian camel breeding strategies, leader males and females might be easily recognized so as to incite the joining of mates when forming and guiding caravans for tourism or at the entrance in the milking parlor.

## 2. Materials and Methods

### 2.1. Study Sample

The study was conducted at the worldwide largest reserve of Canarian camels in Fuerteventura, Canary Islands (28°25′57″ N–14°00′11″ W). The farming environment consisted of square-shaped fenced pens with a shelter providing a shaded area in the middle of the facility and both the feeding and drinking points located along one of the lateral sides.

Four Canarian camel herds were evaluated; the study subjects were all members of these herds. All individuals were recognized by natural markings such moles, scars and fur color patterns. Additionally, the animals were identified with delible numbers placed on the subjects by an operator.

Camel herds were stable (no change, neither introduction nor removal, of any member had been effected prior to the study) and thus selected because intraherd dominance and hierarchy had already been defined and established. Herd structure was as follows: herd one comprised 21 she-camels; herd two, 29 she-camels; herd three, 26 male-camels (20 gelded, 6 entire); and herd four, 24 male-camels (21 gelded, 3 entire). As a result, a total of one hundred Canarian camels (50 cows and 50 male-camels (41 gelded, 9 entire) with average ages (±SD) of 158.36 ± 62.03 months were subjects of the study. The gelded male-camels in the study had been gelded after reaching sexual maturity and after being trained to develop their functional role. Hierarchy within herds was determined following Sueur et al. [1] and Seltmann et al. [28], who reported that the specific organization of individuals during collective movements may constitute the basis for the mechanisms underlying the emergence of complex systems, even if these are not necessarily complex and can be based on local rules, such as hierarchy in a particular herd, and their determinants, such as activity within the herd [29].

### 2.2. A Priori Definitions and Considerations

Provided that the present study has its basis in the determination of motorial selective mimetism, a priori definitions are provided to clarify the manner in which concepts were understood and measurements were therefore taken. As suggested by Seltmann et al. [28], special attention was provided to establishing the conceptual difference between a group movement and any locomotor activity occurring on a daily basis while animals perform their regular activities. In this regard, the following concepts and within-herd roles were predetermined and defined:(a)Initiator/leader: the individual moves directly towards the corridor, where they are restrained for veterinary and other official control activities or duties, and crosses it to a contiguous fenced pen without pausing for more than two seconds. To be considered as an initiation movement, at least two more individuals have to be positioned directly at the entrance of the corridor and just behind the animal crossing it. Three animals are the maximum that can fit, in single file, into the corridor at the same time.(b)Termination: the initiation movement ends when the initiator totally crosses the corridor, enters the contiguous fenced pen, and stops for at least 3 min.(c)Followers: those group members crossing the corridor behind the initiator. They have to arrive at the contiguous fenced pen no later than 3 min after the termination of the movement and approach the initiator at a minimum distance of 3 m.(d)Successful movement: a group movement was considered successful if the initiator had two followers minimum.

### 2.3. Rank Determination 

Once within-herd roles had been defined, intraherd rank was determined. Camels were ranked in a descending order from one to fifty (higher value in the order), such that the camel ascribed the first position (score of 1) was the leader/initiator, while that ascribed with the fiftieth position was the last animal in the hierarchy. Video sampling was used to investigate the types and underlying mechanisms of decision making before and after an individual initiated a movement (became the leader/initiator). Herd movements were recorded using two cameras by two operators (A and B) (Sony RX100M3, 25 fps), with one observer (A) placed on the front of the corridor, the other (B) at the main congregation point of the group. An auxiliary operator (C) annotated the identity of group members conducting predeparture behavior simultaneously (incentive movements or back glances, [30]). Videos recorded lasted for 33.31 ± 32.26 seconds on average (±SD).

An incentive movement was defined as a directed walk of an animal for a distance shorter than an initiation movement that does not result within 2 seconds in feeding, social interactions or lying down.

A back glance was defined as a turn of an individual’s head of more than 90°. Back glances during feeding or social interactions were not considered to be relevant for predeparture behavior and therefore excluded. If the directions of predeparture behaviors formed an angle exceeding 45°, the directions were considered to be different [31].

Once one individual initiated a group movement, one observer (operator B) focused on the initiator and recorded the identity of the initiator, the time of their departure and the identity of followers. Operator C recorded the exact progression order of the joining individuals and the times of their departures. A joiner was defined as an individual moving at an angle of less than 45° to the initiator’s trajectory and crossing an imaginary line situated 4 m (a third of the minimum distance one individual had to move to initiate a group movement) behind the initiator’s start point within 10 min. If the initiator started in the center of the group and individuals ahead of it walked at least 6 m at an angle of less than 45° to the initiator’s trajectory, they were counted as joiners as well. When the initiator returned to the group, the observation was cancelled.

No disruption of the progression order occurred, as the area in which the experiment was conducted was isolated from external influences apart from the animals and the operators conducting the experiment.

Information about dominance relations between individuals was acquired via ad libitum recording of agonistic behavior following the premises described in literature [32,33,34].

No conflict was recorded, as herds had already been conformed prior to the experiments, structure was solid, and herd structure was not distorted, as no new animals were either included or extracted. Although animals were relocated for the experiment, whole herds were relocated to the same testing area at the same time, as literature has suggested [35] that there appears to be little effect of location on animals that are habituated to perform transhumant movements. In this regard, our study considered the findings by authors such as Schulte and Klingel [24], who found that no stable leadership in camels was observed, although individual preferences in the walking order existed when the camels left and entered the enclosure.

### 2.4. Statistical Analysis

#### 2.4.1. Prior Assumption Testing

The Shapiro–Francia W’ test (for 50 < *n* < 2500 samples) and Levene’s test were used to discard gross violations of parametric assumptions (normality and homoscedasticity). The Shapiro–Francia W’ test was performed using the Shapiro–Francia normality routine of the test and distribution graphics package of the Stata Version 16.0 software. Homoscedasticity was tested using Levene’s test with the explore procedure of the descriptive statistics package in SPSS Statistics (Version 25.0, IBM Corp., Armonk, NY, USA) [36].

The chi-square (Χ^2^) test of independence was used to determine the inclusion of variables in the model and to test the probability of the chi-square test on the log ratio, which is equivalent to the Fisher's F test [37], which in turn would test whether every individual had the same probability to be in the analyses. The purpose of the ordinal logistic regression model designed for the present study was to assess conditioning variables that were proximately associated with camel intraherd hierarchy. The variables for which a statistically significant association with camel intraherd hierarchy at a 5% significance level (*p* < 0.05) were used for further analysis using the ordinal logistic regression model. 

#### 2.4.2. Ordinal Logistic Regression

Ordinal logistic regression was used to fit the below statistical model, which describes how the chance of an animal being placed at a specific position within the hierarchy (intraherd hierarchy status) established in camel herds depended on a number of covariates or predictors. We defined *Y* as an ordinal outcome with *J* categories. Thus, we modelled the cumulative probability of responding to a level smaller or equal to j with the probability *P*(*Y* ≤ *j*) for *j* from 1 to the number of categories of *Y*. The analytical expression of the model is as follows: $$logit\left(P\left(Y\leq j\right)\right)=\beta_{j0}+\beta_{j1}x_{1}+\cdots +\beta_{jp}x_{p}$$
for *j* = 1, ⋯, *J* − 1 and p predictors. Because the parallel lines’ assumption, the intercepts were different for each category, but the slopes were constant across categories.

The knowledge of the distribution of intraherd hierarchy status yielded the likelihood of the sample. To estimate the *β* parameters of the model (the coefficients of the linear function), the likelihood function was maximized. As opposed to linear regression, an exact analytical solution does not exist; hence, an iterative algorithm had to be applied.

Maximization of the likelihood function was performed using the Newton–Raphson algorithm with 100 iterations and a convergence level of 0.000001, which are given as default by XLSTAT Version 2014.5.03 [38].

The set of independent covariates and categorical predictors consisted of four blocks: biometrics, phaneroptics, age and sex/sexual status. The first block comprised the variables of height at withers (HW, cm), chest girth (CG, cm), hump girth (HG, cm) and body weight (kg); the second block comprised the variables of coat color, coat particularities (delimited white-haired zones) and eye color; the third block represented the age of the animal and the fourth block comprised the variables of sex and neutering status.

Variables in the first block were chosen because of the implications of overall body condition [39] and body size on the determination of camel intraherd hierarchy [24,40]. The second block was considered because of the implication of phaneroptics with behavioral traits [41]. The third and fourth blocks were included as well, as the variables measured therein have often been reported to be either determinants or confounding in the determination of camel hierarchy status in camels and other species [24,42,43,44]

## 3. Results

### 3.1. Prior Assumption Testing

A gross violation of normality assumption occurred in all variables (*p* < 0.05). Homoscedasticity was violated as well (*p* < 0.01); hence, a nonparametric approach was applied. 

### 3.2. Ordinal Logistic Regression Model

#### 3.2.1. Model Quality

Afterwards, we determined whether the set of variables evaluated in this study may have significantly conditioned (i.e., have been responsible for) intraherd hierarchy status (position of the animals in the hierarchic ranking) by comparing the model as it was defined with a simpler model with only one intercept. In this case, as the probability of these variables modelling for intraherd hierarchy status was lower than 0.001 (Table 1 and Table 2), the variables chosen were concluded to statistically significantly condition and model for intraherd hierarchy status.

Table 2 provides several indicators of the quality of the model (or goodness of fit). These results were equivalent to the R^2^ and to the analysis of variance table in linear regression and ANOVA. The most important value was the probability of the chi-square test on the log ratio. This is equivalent to the Fisher's F test, and it is used to evaluate whether the variables bring significant information by comparing the model as it is defined with a simpler model with only one constant. In this case, as the probability was lower than 0.0001 (Table 1), we could conclude that data could be significantly modelled by the set of variables chosen.

#### 3.2.2. Parameter Analysis

Table 3 provides details on the model and presents a measure of the effect of the variables considered on the categories of the response variable. There is one intercept for each category of the response variable and one set of coefficients, since the parallel curves hypothesis is supposed to be met.

When the regression coefficient for a specific category within a variable was equal to 0.000, this meant that said category was taken as the reference to measure the higher or lower repercussions of the subsequent categories in the same variable. The standardized regression coefficient measured the times that a certain level or category had a higher (positive standardized coefficient) or lower (negative standardized coefficient) repercussion.

The interpretation of parameters was not immediate. Based on the results in Table 3, it was concluded that the model equation for each position n in the intraherd hierarchy status was as follows: 

Log(P(Order ≤ n)/P(Response > n) = 0.050 × HW (cm) – 0.016 × CG (cm) – 0.015 × HG (cm) + 0.001 × Weight (kg) – 0.000 × Age (months) + 3.396 × Chestnut + 6.261 × Bay + 5.820 × Cinnamon + 3.635 × White marks in Extremities, head and neck + 0.669 × Blue Eyes + 5.938 × Gelded.

On this table, we can see from the probability of the chi-squares that the variable most influencing intraherd hierarchy status was whether the animal was gelded or not. The intercept was not significant, but being taller (higher height at withers) and heavier (larger weight), as well as having smaller chests (shorter chest girth) and humps (shorter hump girth), significantly conditioned the animals reaching higher positions in the intraherd hierarchy rank. Age had a very low negative, significant repercussion on intraherd hierarchy status, which means that as an animal grows old, its position in the rank may slightly significantly decrease. Animals with darker coats, such as bay, cinnamon and chestnut; those with white-haired zones in their extremities, head and neck and those with blue eyes were also significantly more prone to reach higher positions in the rank.

## 4. Discussion

According to Cesarani and Pulina [45], genetic selection–domestication induces changes in the behavioral patterns of some farm animals compared to those of their wild ancestors. In this context, hierarchy definition within a herd may be one such behavioral trait, as humans may have reconfigured wild herd structures to make the animals live in artificial groups that may not always resemble those in the wild. The same authors stated that this behavioral domestication process should be considered when planning and implementing farm animal welfare standards at the farm level.

Under the premise of camels being gregarious animals that compulsorily require social behavior expression to ensure their well-being [24], several factors are expected to influence or modulate the establishment and potential temporary modifications of the social structuration within a group of coinhabiting individuals; that is, to influence the trend of eliciting the joining of others when accomplishing a task by providing direction and motivation (leaders) and the relative disposition to associate with one's fellows (submissives).

The moderate R^2^ values obtained (32.8%) may denote that the amplitude of the set of predictive factors involved in the establishment of herd hierarchy may indeed be wider and quite diverse in nature. However, even if other factors that were not registered in this research may condition this type of social hierarchical behavior (leader–follower hierarchy), the variables considered in the analyses explain almost one-third of the variability in intraherd hierarchical status in camels.

### 4.1. Age-Influenced and Sexual Status-Mediated Effects

A slight negative relationship between age (in months) and hierarchy rank position in camels was shown (unstandardized regression coefficient β of −0.007, *p* < 0.05), which was annulled once regression coefficients were standardized. In a previous study with domestic horses, Houpt et al. [46] reached similar conclusions, as age appeared to influence neither agonistic behavior nor social structure within the herd. For companion dogs, Pal et al. [47] reported that within-group hierarchy was also not correlated to age. In general terms, age seems not to be a major influencing factor for group hierarchy in domestic animals, although it may be linked to other cognitive features or processes that may make handling easier or more effective, such as cognitive bias [48,49]. In this regard, age could be related to certain aptitudes or behaviors derived from the cognitive development and social needs of the animals [50], similarly as in humans [51]. Additionally, it can be assumed that both the greater physical vigor and stamina levels and the lower fearfulness associated with younger age in gregarious animal species make younger animals more likely to be endorsed as leaders. 

Exploratory activities and other energetic tasks are more frequent at an early age and help in developing skills that involve the use of information independent of acquired knowledge and the demonstration of creativity to solve problems in novel situations (‘fluid intelligence’) [52]. Furthermore, the need to relate with others is greater in younger individuals, which makes them engage in more relationship-oriented activities than older congeners [53], which in turn increases the individual probability of being followed by congeners depending on the social status. Empirical examples in various animal species demonstrate such increased prevalence of high-energy behaviors in younger compared to older subjects [54,55].

This has also been reported for free-living animal species and humans, as several researchers [9,56,57] have suggested that an increase in age is associated with a greater accumulation of knowledge or experiences that help group decision making and survival. In a closely related phenomenon, herd size has been observed to be generally larger in animals living in open habitats such as savannas and pastures, as better territorial and in-group defensive behavior is thereby enabled [58].

A general explanation for this differential influence of age in leadership hierarchy could rely on the basis that animals in domestic scenarios do not need to engage in alert reactions towards external threats, since they are reared in relatively highly controlled environments. In this context, group size may not be important for survival reasons, but it may be important for animal welfare issues arising from confined housing [59]. This reinforces the hypothesis that age may not be of critical importance from the point of view of knowledge accumulation, because it is assumed that all animals in the same herd are exposed to the same stimuli and that the age range within each group is always kept as homogeneous as possible for ease of animal handling and technical efficiency for caregivers. More than sex, the sexual status of the animals was reported to influence intraherd hierarchy position. The maturation and possible temporary or permanent functional alterations of the endocrine system plays a fundamental role in the modulation of animal behavior and could, together with age and cognitive development, explain the results discussed so far alongside the implications of castration and the time of its performance on animal temperament. According to our results, gelded animals were more likely to lead group movements or collective activities. The maintenance of adequate levels of serotonin trough the active interaction with the environment (curiosity) strengthens the antagonistic potential of this neurotransmitter on the effects of sex hormones at the central level (prefrontal cortex and subcortical structures) [60]. Consequently, agonistic behaviors such as aggression or sexual-related behaviors are reduced [61,62]. According to the common practice of castration in this breed, the animals are castrated once they have been initiated in the domestication protocol for their functional aptitude and have reached their sexual maturity, having reached sufficient serum levels of sex hormones to ensure proper general organic maturity (brain structures, among others) but not a harmful level from an ethological perspective [63]. Such conditions may arise as age progresses, and therefore, not gelded animals are more territorial, their social circles are more restricted and the frequency of their social interaction is minimized. Furthermore, the older an animal is when castration is performed, the lower the probability is of unwanted behaviors associated with the intrinsic activity of sex hormones disappearing, because these behaviors already constitute fixed patterns in the behavioral repertoire of the individual [64].

Additionally, the castration of camels has been shown to have a great impact on body development in this species, which constitutes a further criterion of functional interest complementary to behavioral traits in working animals. Specifically, some authors [65,66] have discussed the advantages of castration, if carried out after the animal is sexually mature, in camels relegated to working activities. These animals were found to be larger, more robust [67] and more enduring [68] under the effect of postneutering benefits in the performance of working animals. In this context, Wilson [69] also added that, in the case of males, castration promoted a greater development of the soft palate. This anatomical structure, when voluntarily protruded outwards, could play an important role in the definition of individual social status at the intraherd level or when displaying defense against invasion by foreigners.

### 4.2. Leadership Inference from Physical External Appearance

A direct connection can be drawn between the practical framework set out in the preceding paragraph and the small positive influence of body weight and the probability of becoming a leader that was revealed in our data. This finding adds more evidence to the existing literature of the zoometric variables that influence the leadership hierarchy of camels and that are in turn influenced by management practices. In this regard, more than large animals (tall, with a wide chest and big hump), which balance to negative regression coefficients, heavier animals may play a determinant role in leadership and hierarchy definition. An empirical association has been described between body weight and sexual status (neutered or non-neutered) for housed domestic horses [70], outdoor-living domestic horses [71] and female chamois [72]. Hence, it can be concluded that not only the individual genetic background (and other factors such as diet) but castration may have a positive influence on body development. As long as castration is practiced properly, positive impacts on the degree of organic development and integral functional performing (physical resistance and behavior) of camels could arise. By applying this rationale to management programs, the efficiency of handling practices can be substantially improved. This may translate into tangible benefits for both staff and animals as well as consumers participating in interactive activities with the animals (e.g., camelback riding tours).

By contrast, the aforementioned negative regression coefficient for the variables of HW, CG and HG (and age before standardization) may indeed be determinant for the individual probability of leading collective actions. This finding may be of valuable help at the time of designing facilities and defining herd management protocols. For instance, it may aid in constructing proper dimensions for the entrance circuits to the milking parlors for females and in deciding which females would be the best candidates to lead the group at these emplacements. When handling males, camel herders would be able to select animals that are preferred to lead the caravans in order to prevent disruptions due to fearfulness or mistrustfulness if they encounter obstacles along the routes.

### 4.3. Coat and Eye Color Genetics May Reflect Camel Temperament

A wealth of information can be found in the scientific literature on the pleiotropic effects of the genes responsible for phaneroptical characters such as coat and eye color on the development and function of neural structures [73,74,75]. In this context, temperament features such as calmness and nervousness were reported to be quantitatively differentiated on the basis of coat color among individuals of the same species [76,77,78,79] and thus further drive the selection criteria for breeding purposes.

For example, Finn et al. [80] found that chestnut horses were more likely to approach novel objects and animals. This finding may evidence a direct consequence of domestication and bias selection towards boldness, since the bay phenotype was more prevalent prior to the species’s domestication. Parallelly, horses with the Silver mutation Arg618Cys in the PMEL gene were more cautious when being presented novel stimuli or approaching novel objects [74]. Applied research in domestic dogs showed that individuals presenting white coat color were more fearful and displayed more submissive reactions [81]. In mountain sheep [82] and lions [83], social rank was higher as coat darkened.

For the particular case of camels, Almathen et al. [84] found a significant association between the polymorphisms in the MC1R and ASIP genes and variability in coat color in this species. However, these authors did not refer to associated behavioral changes, although it was assumed that they existed, following the conclusions found for other species. Globally, applied research in this field has suggested that the magnitude of the aforementioned pleiotropic effects appears to be greater in dark-colored animals [85] and that light- and particolored animals also frequently suffer from congenital deafness [41] and ocular anomalies [74,86,87] due to gene mutations in KIT [88], which affect training.

It is important to highlight the common perception of camel herders that animals with a variable proportion of white fur (piebaldness) are the least aggressive but also the most fearful and submissive [89], which impairs their individual ability to lead collective actions. The same conclusion has been reached in dogs [78,90] and foxes [91] in domestic settings.

The extension and distribution of white spots conditions the degree of deafness and visual deficit. Both impairments have been reported to be associated to certain predominant behavioral patterns [41]. This has been specifically dealt with in other species such as cows. In this regard, Grandin et al. [92] reported the level of depigmentation to be a strong driving agent of the dimension of impairments and consequently of the associated behavioral patterns. For instance, Holstein cows with complete depigmented white areas on their heads are among the calmest, while those that are mostly white on the body are nervous and intractable, which was supported by our study, as camels presenting white spots on their extremities, head and neck significantly scored relatively lower in the hierarchical rank.

This depigmentation throughout the body has also relatively frequently been reported to be associated with iris depigmentation and heterochromia in camels mainly reared across northwestern African countries and the Canary Islands [41]. Our results suggested that blue eyed animals scored relatively lower in hierarchic rank. As it occurs in depigmented coated animals, eye color has been related to differences in reactivity to external stimuli (‘eye color–reactivity hypothesis’), whether such stimuli are familiar or not. Pastoralists often regard piebald camels as reckless, stubborn and disobedient or particularly tame, even numb. Indeed, several authors have agreed that subjects with darker eyes tend to display greater reactive skills [93] and a superior speed of locomotion [94,95]. In this context, Sahrawi herders describe piebald camels as having different levels of deafness, with increased deafness linked to the presence of blue eyes and white coloring of the head and toes. However, others, such as bold pie camels (black ears and nails) have normal hearing [96].

Authors such as Volpato et al. [41] transcribed the widespread knowledge among Sahrawi and Tuareg herders, who related complete deafness to calm behavior while contrastingly ascribing partial deafness conditions to increasingly agitated camels with unpredictable reactions (stubborn, disobedient and reluctant to understand orders). Additionally, these authors reported piebald animals to achieve a quieter and tamer secondary position in status within the herd [97]. According to Volpato et al. [41], the basis for these animals being ranked at relatively lower positions in the intraherd hierarchy lay upon their bad night eyesight and their increased likelihood to get lost when light is low, which translated into some piebald male-camels’ lower ability to manage the herd.

## 5. Conclusions

Although intraherd hierarchy may indeed be driven by a wide and diverse set of etiological factors, phaneroptics and zoometry may play a remarkable role. More than age, the sexual status of the animal (entire or castrated) influenced intraherd hierarchy position as the maturation and possible temporary or permanent functional alterations of the endocrine system play a fundamental role in the modulation of animal behavior and brain development in this species. More than large animals (tall, with a wide chest and big hump), heavier individuals may play a determinant role in leadership and hierarchy definition. Dark-coated camels scored higher in the hierarchical rank than those presenting light coats or larger extensions of white all over the body (extremities, head and neck). The basis for these animals being ranked at lower positions in the intraherd hierarchy may lie in visual and acoustic impairments, which make them prone to develop a limited ability to manage the herd. The information obtained in this study is helpful for routine intraherd management and for the genetic management of herds with the aim to define and preselect potential leaders, which is of prominent importance for the touristic application of the representatives of this breed.

## Figures and Tables

**Table 1 animals-11-02886-t001:** Test of the null hypothesis H0: Y = 0 (variable: intraherd hierarchy status).

Statistic	DF	Chi-Square	Pr > Chi^2^
−2 Log(Likelihood)	19	39.753	0.004
Score	19	41.264	0.002
Wald	19	35.518	0.012

**Table 2 animals-11-02886-t002:** Goodness of fit statistics for intraherd hierarchy status model.

Statistic	Full Model
Observations	100
Sum of weights	100
Df	32
−2 Log (Likelihood)	720.675
R^2^ (McFadden)	0.052
R^2^ (Cox and Snell)	0.328
R^2^ (Nagelkerke)	0.328
AIC	856.675
SBC/BIC	857.173
Iterations	6

Df: degrees of freedom; AIC: Akaike’s Information Criterion; SBC/BIC: Schwarz’s Bayesian Criterion/Bayesian Information Criterion.

**Table 3 animals-11-02886-t003:** Standardized regression coefficients for the factors and covariates considered in the model for the intraherd hierarchy status.

Source	Hierarchy Categories	Standardized Regression Coefficient (β)	Standard Error	Wald Chi-Square	Pr > Wald Χ^2^	Wald Lower Bound (95%)	Wald Upper Bound (95%)
Intercept	1	63.212	44.324	2.034	0.154	−23.662	150.087
	2	62.437	44.313	1.985	0.159	−24.415	149.289
	3	61.965	44.306	1.956	0.162	−24.873	148.802
	4	61.613	44.300	1.934	0.164	−25.213	148.440
	5	61.325	44.295	1.917	0.166	−25.491	148.142
	6	61.072	44.290	1.901	0.168	−25.734	147.878
	7	60.848	44.285	1.888	0.169	−25.949	147.645
	8	60.650	44.283	1.876	0.171	−26.142	147.442
	9	60.466	44.282	1.865	0.172	−26.325	147.257
	10	60.287	44.281	1.854	0.173	−26.502	147.076
	11	60.110	44.278	1.843	0.175	−26.674	146.894
	12	59.939	44.275	1.833	0.176	−26.839	146.717
	13	59.773	44.273	1.823	0.177	−27.001	146.546
	14	59.608	44.271	1.813	0.178	−27.161	146.377
	15	59.443	44.269	1.803	0.179	−27.322	146.208
	16	59.277	44.268	1.793	0.181	−27.487	146.042
	17	59.114	44.268	1.783	0.182	−27.651	145.878
	18	58.949	44.267	1.773	0.183	−27.813	145.711
	19	58.785	44.265	1.764	0.184	−27.973	145.543
	20	58.619	44.264	1.754	0.185	−28.136	145.374
	21	58.445	44.262	1.744	0.187	−28.308	145.198
	22	58.326	44.261	1.737	0.188	−28.424	145.077
	23	58.206	44.260	1.729	0.188	−28.542	144.955
	24	58.085	44.259	1.722	0.189	−28.662	144.831
	25	57.959	44.258	1.715	0.190	−28.786	144.703
	26	57.828	44.256	1.707	0.191	−28.913	144.569
	27	57.689	44.254	1.699	0.192	−29.048	144.425
	28	57.539	44.251	1.691	0.194	−29.191	144.270
	29	57.379	44.248	1.682	0.195	−29.345	144.102
	30	57.295	44.246	1.677	0.195	−29.425	144.015
	31	57.211	44.245	1.672	0.196	−29.508	143.931
	32	57.129	44.246	1.667	0.197	−29.592	143.849
	33	57.046	44.247	1.662	0.197	−29.678	143.769
	34	56.961	44.248	1.657	0.198	−29.764	143.686
	35	56.871	44.250	1.652	0.199	−29.856	143.599
	36	56.778	44.251	1.646	0.199	−29.952	143.509
	37	56.680	44.253	1.641	0.200	−30.053	143.414
	38	56.577	44.254	1.634	0.201	−30.160	143.313
	39	56.467	44.256	1.628	0.202	−30.273	143.207
	40	56.348	44.257	1.621	0.203	−30.394	143.090
	41	56.218	44.258	1.614	0.204	−30.525	142.961
	42	56.075	44.258	1.605	0.205	−30.669	142.819
	43	55.915	44.258	1.596	0.206	−30.829	142.659
	44	55.740	44.258	1.586	0.208	−31.003	142.483
	45	55.545	44.257	1.575	0.209	−31.198	142.288
	46	55.313	44.258	1.562	0.211	−31.430	142.057
	47	55.015	44.258	1.545	0.214	−31.729	141.759
	48	54.597	44.260	1.522	0.217	−32.151	141.344
	49	53.893	44.266	1.482	0.223	−32.867	140.652
Animal inherent	HW (cm)	−0.050	0.016	9.452	0.002	−0.082	−0.018
	CG (cm)	−0.016	0.010	2.646	0.104	−0.035	0.003
	HG (cm)	−0.015	0.005	8.186	0.004	−0.026	−0.005
	Weight (kg)	0.001	0.000	8.087	0.004	0.000	0.002
	Age (months)	0.000	0.000	8.976	0.003	0.000	0.000
Sex	Female	0.000	0.000				
	Male	1.233	1.675	0.542	0.462	−2.050	4.517
Coat color	Roan	0.000	0.000				
	Chestnut	3.396	1.729	3.858	0.050	0.007	6.785
	Bay	6.261	2.545	6.052	0.014	1.273	11.249
	Cinnamon	5.820	1.948	8.922	0.003	2.001	9.639
	Blonde	4.315	2.869	2.263	0.132	−1.307	9.938
	Black	8.267	11.400	0.526	0.468	−14.076	30.610
	White	−0.415	15.652	0.001	0.979	−31.094	30.263
Coat particularities (delimited white-haired zones)	All over	0.000	0.000				
	Extremities	2.012	2.226	0.817	0.366	−2.351	6.375
	Extremities, head and neck	3.635	1.692	4.616	0.032	0.319	6.951
	Solid color (no white)	2.125	2.231	0.907	0.341	−2.248	6.497
	Head and neck	12.766	7.532	2.873	0.090	−1.996	27.529
Eye color	Brown	0.000	0.000				
	Blue	6.669	2.936	5.159	0.023	0.914	12.423
	Brownish with blue spots	2.158	13.531	0.025	0.873	−24.363	28.678
Sex status	Whole	0.000	0.000				
	Gelded	5.938	1.883	9.948	0.002	2.248	9.627

## Data Availability

The data presented in this study are available on request from the corresponding author.

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
