# Peer review of "The Youngest, the Heaviest and/or the Darkest? Selection Potentialities and Determinants of Leadership in Canarian Dromedary Camels"

_animals, 2021, doi:10.3390/ani11102886_

Round 1

Reviewer 1 Report

This is an interesting paper.  The analysis seems appropriate and interpretation fair.  I am not an expert on the statistical approach, but it appears to be valid. I recommend publishing with some corrections to grammar as I note in the manuscript.

Author Response

Response: We thank the reviewer for his/her kind comments. Each comment addressed by reviewer in the pdf file was applied.

Reviewer 2 Report

Dear authors,

the manuscript entitled “The youngest, the heaviest and/or the darkest? Selection potentialities and determinants of leadership in Canarian dromedary camels” deals with the study of the possibilities to perform genetic selection in camels.

The manuscript is nicely written, and the aim is of interest. It is probably the first time that I could not find any problem in one manuscript.

One author, Elena Ciani, is missing in the list of authors in the website.

Just one comment: it seems that modern livestock breeds were “forced” by human to live in bigger groups and, therefore, also the hierarchy among animals could have been influenced by humans. Recently, a review about behavior and welfare in livestock has been published in this same journal (10.3390/ani11030724): you can probably find there something interesting for the discussion. Those authors stated that the behavior of farmed animals has been changed during domestication and animal breeding processes.

Few minor line-by-line comments:

Lines 109-111: “All individuals were recognized by natural markings like moles, scars and fur color patterns but also the animals were identified with delible numbers placed on the subjects by an operator” please consider replacing “… but also the animals...” with “…but also they were…” since the sentence has already all individuals at the beginning.

Line 114: “Herd structure is as follows…” should be “Herd structure was as follows…” since all the other verbs are in the past form.

Line 128: please consider rephrasing this sentence.

Line 247: please check this sentence, in particular “… may significantly condition”

Author Response

Reviewer 2

Dear authors,

the manuscript entitled “The youngest, the heaviest and/or the darkest? Selection potentialities and determinants of leadership in Canarian dromedary camels” deals with the study of the possibilities to perform genetic selection in camels.

The manuscript is nicely written, and the aim is of interest. It is probably the first time that I could not find any problem in one manuscript.

Response: We thank the reviewer for his/her kind comments.

One author, Elena Ciani, is missing in the list of authors in the website.

Response: We amended it.

Just one comment: it seems that modern livestock breeds were “forced” by human to live in bigger groups and, therefore, also the hierarchy among animals could have been influenced by humans. Recently, a review about behavior and welfare in livestock has been published in this same journal (10.3390/ani11030724): you can probably find there something interesting for the discussion. Those authors stated that the behavior of farmed animals has been changed during domestication and animal breeding processes.

Response: We considered the suggestion of the reviewer and included the information and citation in the discussion.

Few minor line-by-line comments:

Lines 109-111: “All individuals were recognized by natural markings like moles, scars and fur color patterns but also the animals were identified with delible numbers placed on the subjects by an operator” please consider replacing “… but also the animals...” with “…but also they were…” since the sentence has already all individuals at the beginning.

Response: Rewritten.

Line 114: “Herd structure is as follows…” should be “Herd structure was as follows…” since all the other verbs are in the past form.

Response: Changed.

Line 128: please consider rephrasing this sentence.

Response: Rephrased.

Line 247: please check this sentence, in particular “… may significantly condition”

Response: We revised it and clarified it.